# Burden of soil-transmitted helminth infection in pregnant refugees and migrants on the Thailand-Myanmar border: Results from a retrospective cohort

Tobias Brummaier[1,2,3,4]*, Nay Win Tun[1], Aung Myat Min[1,2], Mary Ellen Gilder[1,5], Laypaw Archasuksan[1], Stephane Proux[1], Douwe Kiestra[1], Prakaykaew Charunwatthana[6,7], Jürg Utzinger[3,4], Daniel H. Paris[3,4], Mathieu Nacher[8,9], Julie A. Simpson[10], Francois Nosten[1,2], Rose McGready[1,2]

1 Shoklo Malaria Research Unit, Mahidol-Oxford Tropical Medicine Research Unit, Faculty of Tropical Medicine, Mahidol University, Mae Sot, Thailand, 2 Centre for Tropical Medicine and Global Health, Nuffield Department of Medicine, University of Oxford, Oxford, United Kingdom, 3 Swiss Tropical and Public Health Institute, Basel, Switzerland, 4 University of Basel, Basel, Switzerland, 5 Department of Family Medicine, Faculty of Medicine, Chiang Mai University, Chiang Mai, Thailand, 6 Mahidol-Oxford Tropical Medicine Research Unit, Faculty of Tropical Medicine, Mahidol University, Bangkok, Thailand, 7 Department of Clinical Tropical Medicine, Faculty of Tropical Medicine, Mahidol University, Bangkok, Thailand, 8 Centre d'Investigation Clinique Antilles Guyane, Centre Hospitalier Andree Rosemon Cayenne, French Guiana, 9 Département Formation Recherche Santé, Université de Guyane, Cayenne, French Guiana, 10 Centre for Epidemiology and Biostatistics, Melbourne School of Population and Global Health, University of Melbourne, Melbourne, Australia

* tobias@shoklo-unit.com

**Data Availability Statement:** Data cannot be shared publicly because this is a population of undocumented refugees and migrants. Data are

## Abstract

### Background

Soil-transmitted helminth (STH) infections are widespread in tropical and subtropical regions. While many STH infections are asymptomatic, vulnerable populations such as pregnant women face repercussions such as aggravation of maternal anaemia. However, data on prevalence and the effect of STH infections in pregnancy are limited. The aim of this analysis was to describe the burden of STH infections within and between populations of pregnant women from a local refugee camp to a mobile migrant population, and to explore possible associations between STH infection and pregnancy outcomes.

### Methodology

This is a retrospective review of records from pregnant refugee and migrant women who attended Shoklo Malaria Research Unit antenatal care (ANC) clinics along the Thailand-Myanmar border between July 2013 and December 2017. Inclusion was based on provision of a stool sample during routine antenatal screening. A semi-quantitative formalin concentration method was employed for examination of faecal samples. The associations between STH mono-infections and maternal anaemia and pregnancy outcomes (i.e., miscarriage, stillbirth, preterm birth, and small for gestational age) were estimated using regression analysis.

available from the Mahidol-Oxford Research Unit Institutional data access committee (contact Rita Chanviriyavuth, email: rita@tropmedres.ac) for researchers who meet the criteria for access to confidential data.

**Funding:** The authors received no specific funding for this work.

**Competing interests:** The authors have declared that no competing interests exist.

## Principal findings

Overall, 12,742 pregnant women were included, of whom 2,702 (21.2%) had a confirmed infection with either *Ascaris lumbricoides*, hookworm, *Trichuris trichiura*, or a combination of these. The occurrence of STH infections in the refugee population (30.8%; 1,246/4,041) was higher than in the migrant population (16.7%; 1,456/8,701). *A. lumbricoides* was the predominant STH species in refugees and hookworm in migrants. *A. lumbricoides* and hookworm infection were associated with maternal anaemia at the first ANC consultation with adjusted odds ratios of 1.37 (95% confidence interval (CI) 1.08–1.72) and 1.65 (95% CI 1.19–2.24), respectively. Pregnant women with *A. lumbricoides* infection were less likely to miscarry when compared to women with negative stool samples (adjusted hazard ratio 0.63, 95% CI 0.48–0.84). STH infections were not significantly associated with stillbirth, pre-term birth or being born too small for gestational age. One in five pregnant women in this cohort had STH infection. Association of STH infection with maternal anaemia, in particular in the event of late ANC enrolment, underlines the importance of early detection and treatment of STH infection. A potential protective effect of *A. lumbricoides* infection on miscarriage needs confirmation in prospective studies.

## Author summary

Soil-transmitted helminths (STHs) are parasitic worms transmitted through the faecal-oral or transcutaneous route. STHs are ubiquitous in tropical and subtropical regions where access to clean water, improved sanitation, and hygiene are lacking. STH infections are often asymptomatic; however, heavy infections can cause adverse effects such as abdominal discomfort and anaemia. Anaemia in pregnancy can increase the risk of pregnancy problems such as miscarriage, preterm delivery, and poor fetal growth. This retrospective analysis of routinely collected data from more than 12,000 pregnant refugee and migrant women who attended antenatal care clinics along the Thai-Myanmar border shows that one in five pregnant women were infected with STHs. Moreover, pregnant women with a STH infection were more likely to present with anaemia, suggesting that early presentation to antenatal care as well as anthelmintic treatment with subsequent anaemia prophylaxis may reduce the risk of anaemia. Interestingly, our data also indicate that STH infections may have beneficial effects on pregnancy outcomes, as miscarriage was less likely to occur in pregnancies with maternal STH infections. This finding was significant for infection with the roundworm *Ascaris lumbricoides*. Verification of these findings in prospective studies is warranted.

## Introduction

Estimates from the World Health Organization (WHO) suggest that approximately 25% of the world's population is infected with soil-transmitted helminths (STHs) [1], causing a global burden of 1.92 million disability-adjusted life years (DALYs) [2]. The common STH species are *Ascaris lumbricoides*, *Trichuris trichiura*, and the two hookworms *Ancylostoma duodenale* and *Necator americanus*. Most STH infections occur in tropical and subtropical regions, and disproportionally affect resource-limited settings where the population faces barriers in access to clean water, adequate sanitation, and general hygiene [3,4]. STH infections are often

asymptomatic, but can present with asthenia, loss of appetite, and variable abdominal symptoms [3]. Intestinal helminthiasis has repercussions on nutritional uptake leading to reduced food intake, impaired digestion, and malabsorption [5]. The latter predominantly leads to unfavourable consequences, such as poor growth or development in children and depletion of iron stores with subsequent anaemia [5]. Maternal anaemia is a public health issue in many regions, often aggravated by STH infection and associated with unfavourable pregnancy outcomes [3,6,7]. Chronic blood loss and mucosal inflammation caused by intestinal nematode infections increase with intensity of infection and explain the association between hookworm infection and anaemia in pregnancy and, to a smaller extent, *T. trichiura* infection [8–10].

Based on population and STH endemicity estimates, approximately 688 million women of reproductive age (i.e., 15–49 years) require preventive chemotherapy for STH worldwide and the effect of STH infection on pregnancy needs attention [11]. Recent data suggest that even though only 20.5% of the women requiring treatment globally were treated in 2015, about 622,000 DALYs were averted by providing this intervention [11,12]. The positive effects of anthelmintic treatment during pregnancy are mostly attributed to an increase in maternal haemoglobin and were observed when concomitant iron supplementation was provided [13–15]. Conversely, positive effects of *A. lumbricoides* have been reported; for instance, *A. lumbricoides* creates a favourable immunological environment for the fetus reflected by increased fecundity [16]. However, little is known about the effects of STH treatment during pregnancy on maternal and neonatal morbidity [17].

The aim of this study was to assess the burden of STH infection in a pregnant population living on the Thailand-Myanmar border with no routine chemoprevention (deworming) and to describe the prevalence of different STH species in a mobile migrant population and a population confined to the largest refugee camp along the Thailand-Myanmar border. Additionally, we report an exploratory analysis on associations between STH infection, maternal health, and pregnancy outcomes.

## Methods

### Ethics statement

Retrospective analysis of routinely collected antenatal care (ANC) data was approved by the Ethics Committee of the Faculty of Tropical Medicine at Mahidol University (Ethics reference: TMEC 17–027). Note that attendance at ANC is voluntary. All women are assigned to a group and one-on-one counselling in relation to screening tests that are offered routinely at Shoklo Malaria Research Unit (SMRU). All women have the choice to opt-out of the screening procedures.

### Study setting and population

This analysis of prospectively and routinely collected ANC data included pregnant women from a refugee and mobile migrant population residing on the Thailand-Myanmar border. The region is a rural and tropical environment with a high STH burden as previously reported [18]. SMRU is a field station of the Faculty of Tropical Medicine, Mahidol University (Bangkok, Thailand), and is part of the Mahidol-Oxford Research Unit. SMRU combines humanitarian work and clinically relevant research with a focus on maternal and child health. All SMRU consultations are free of charge and registration to the ANC program is voluntary. Pregnant women were either followed in clinics at one of two sites for migrants (Mawker Thai (MKT) and Wang Pha (WPA)), or in Mae La (MLA) refugee camp (Fig 1).

According to the United Nations High Commissioner for Refugees (UNHCR), the average population size of MLA over the study period was 41,700 refugees living in an area of approximately 2 km$^2$ (population density 20,850/km$^2$). In this population-dense environment, non-

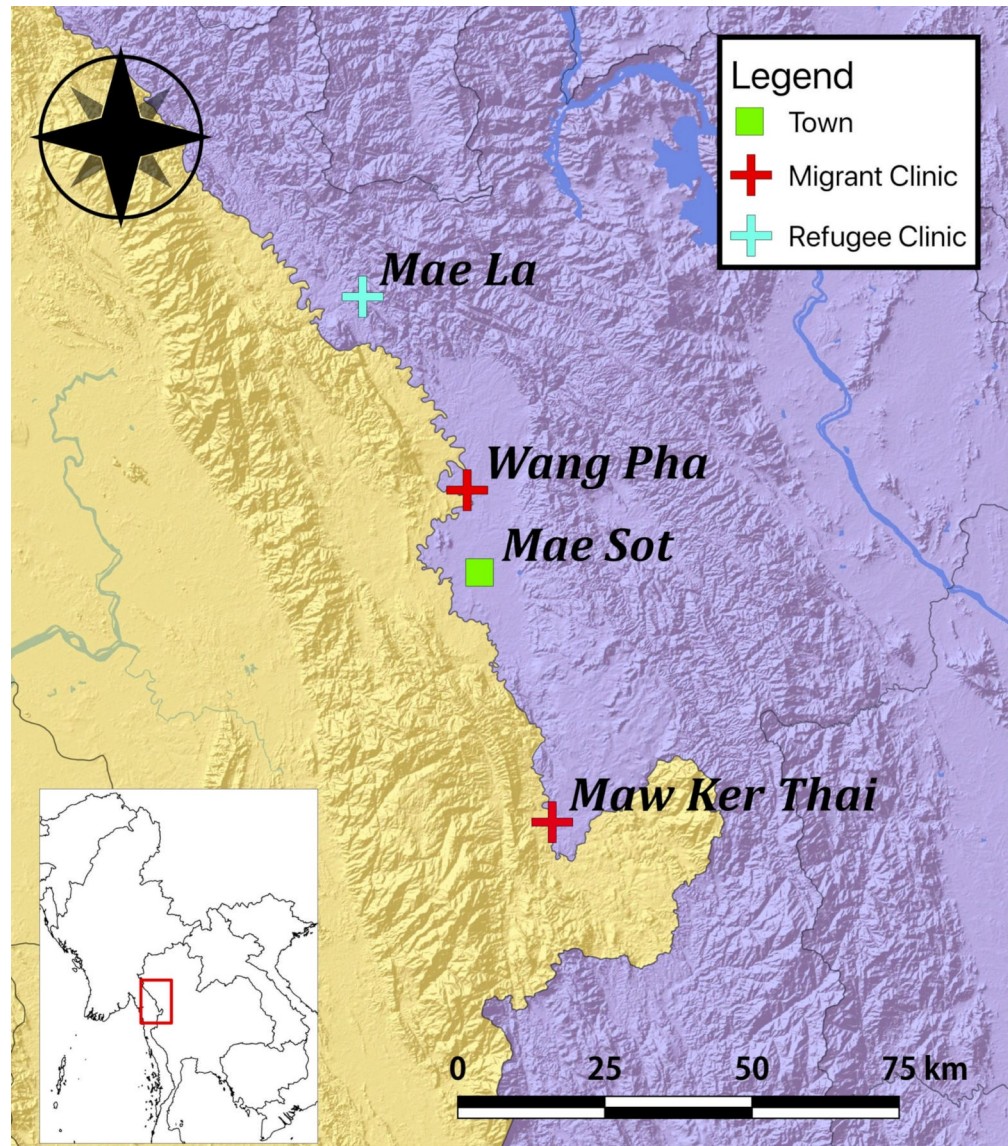

**Fig 1. Location of Shoklo Malaria Research Unit antenatal care clinics on the Thailand-Myanmar border.**

governmental organizations (NGOs) support water (wells with pumps) and sanitation (predominantly pit toilets) for displaced persons from Myanmar, mostly of Karen ethnicity. School-age children in MLA camp receive mebendazole deworming on a 6-monthly basis from medical NGOs. Outside the camps, migrants tend to be mobile, so housing is often temporary, the quality of water and sanitation is variable (often poor), and deworming is not routinely available.

Data from records of all women enrolled in the SMRU ANC program between July 1, 2013 and December 31, 2017 were analysed. Pregnant women who did not provide a stool sample at ANC, multiple (twin) pregnancies and hydatidiform moles were excluded from the analysis (S1 Table).

As per routine ANC procedures, the gestational age was estimated by obstetric ultrasound at the first visit. All three sites offer 24-hour birthing services with skilled birth attendants and

emergency referral for caesarean section. More than 80% of women deliver at SMRU where the neonate is weighed on an electronic SECA 354 scale (precision 5 g) with weekly calibration.

Routine anaemia and malaria detection was conducted through active screening: malaria blood films (every 2 weeks in this unstable transmission setting) and haematocrit (HCT; approximately 5 times per pregnancy at first ANC, 12–16, 22, 28, and 34–36 weeks) [19]. Malaria treatment was provided in accordance with WHO guidelines, anaemia prophylaxis was offered at each visit (ferrous sulphate 200 mg/day, equivalent to 60 mg elemental iron, folate 5 mg per week) or treatment given (ferrous sulphate 400 mg twice daily, vitamin C 100 mg twice daily, folate 5 mg per day), if anaemia was detected. Complete blood counts (CBC) were done routinely for all women at their first ANC contact until the end of 2016. CBCs done 28 days prior or at the same day as the stool sample, were included in the analysis of the association between eosinophils and STH infections.

Routine screening for intestinal parasite infections for all pregnant women was introduced in mid-2013 by providing a fresh faecal sample at the first ANC visit. In case pregnant women were unable to provide a stool sample during the first visit, a small plastic container was given to each woman to collect a sample at home on the morning of the next ANC visit. In the event of a STH-positive sample, anthelmintic treatment was provided: for hookworm and *A. lumbricoides*, either using albendazole (single oral dose of 400 mg) or mebendazole (100 mg twice daily for 3 days), and for *T. trichiura* either albendazole (400 mg once daily for 3 days) or mebendazole (100 mg twice daily for 3 days). Treatment of first trimester infections was delayed until 14 weeks of gestation.

## Definitions

Status was determined by residence: refugees from MLA camp or migrants from MKT or WPA. Definitions of ethnicity followed a self-identification approach with women nominating their own ethnic affiliation, using the locally preferred term [20].

Anaemia was defined as any HCT level less than 30%. Relative eosinophilia was defined as $\geq$6% of circulating leucocytes [21]. Body-mass index (BMI) definitions followed recommendations for Asian BMI groups [22]: underweight <18.5 kg/m$^2$; normal weight 18.5 to <23 kg/m$^2$; overweight 23 to <27.5 kg/m$^2$; obese $\geq$27.5 kg/m$^2$. Ideally, BMI assessment is based on the preconception BMI but as this was not known for most women, a BMI of <18.5 kg/m$^2$, at any stage of pregnancy, was used instead.

Miscarriage was defined as fetal loss before 28$^{+0}$ weeks of gestation; cases for the assessment of miscarriage were only considered if the first ANC contact was in the first trimester (estimated gestational age (EGA) less than 14 weeks). Stillbirth was defined as birth of a fetus with no signs of life, at a gestation of $\geq$28$^{+0}$ weeks. SMRUs definition for stillbirth follows that recommended by WHO for international comparison and the rationale for the miscarriage definition, which has no agreed international standard, is provided elsewhere [23]. Preterm birth (PTB) was defined as pregnancy outcome between 28$^{+0}$ and 37$^{+0}$ weeks of gestation; only cases with their first ANC contact before completion of 37 weeks of gestation were considered for this analysis. Small for gestational age (SGA) was defined as a birthweight (within 72 hours) below the 10$^{th}$ centile, following gestation and sex adjusted birthweight centiles as published by the Intergrowth 21$^{st}$ Consortium [24].

## Diagnosis of STH infection

All faecal samples were processed in the on-site laboratory following standard procedures for sample preparation and interpretation. A formalin concentration method was used to examine

faecal samples. Briefly, 500 mg of faecal matter was mixed with 10 ml of 10% formol saline and homogenized until all faecal material was suspended. This emulsion was filtered through a moulded strainer (Caplugs Evergreen FPC Fecal Parasite Concentrator; Rancho Dominguez, California, United States of America), whose 0.6 mm x 0.6 mm holes allow parasite eggs to pass, while excess faecal debris are retained. After centrifugation (500 $g$ for 5 min), the super-natant was discarded, and the remaining deposit was re-suspended in a 0.85% saline solution. Finally, wet mounts with 1–2 drops of this solution were prepared, and slides were examined under a microscope by experienced laboratory technicians. In STH-positive samples, a semi-quantitative approach to estimate the intensity of infection was adapted. Intensity of infection was categorized by counting the number of helminth eggs per slide into rare (1 egg/slide), low (2–3 eggs/slide), medium (4–10 eggs/slide), and high (>10 eggs/slide).

## Statistical analysis

Demographic characteristics of this cohort were described by median (inter-quartile range (IQR)) and frequency (proportion), where appropriate. Logistic regression modelling was performed to assess the association between anaemia at the first ANC contact and STH infection. The model was adjusted for the following potential confounders that were selected based on content knowledge: (i) STH categories, (ii) migration status, (iii) smoking, (iv) age, (v) gravidity (primi- vs multigravida), (vi), trimester first ANC contact, (vii), underweight, (viii) malaria 1st ANC contact.

To estimate the association between STH infection and pregnancy outcomes, Cox proportional hazards models were fitted for the outcome miscarriage and stillbirth, and logistic regression models for the outcomes PTB and SGA.

The models were adjusted for the following confounders:

1. Association of STH infections and miscarriage: (i) STH categories, (ii) migration status, (iii) smoking, (iv) age, (v) gravidity, (vi) history of miscarriage, (vii) ethnicity, (viii) literacy status, (ix) anaemia at first HCT and (x) non-malarial fever 1st trimester.

2. Association between STHs and stillbirth: (i) STH categories, (ii) migration status, (iii) smoking, (iv) age, (v) gravidity, (vi) history of stillbirth, (vii) literacy status, (viii) anaemia in pregnancy.

3. Association between STH infections and preterm birth: (i) STH categories, (ii) migration status, (iii) smoking, (iv) ethnicity, (v) age, (vi) gravidity (primigravida vs multigravida), (vii) previous PTB, (viii) underweight, (ix) GH/Chronic HTN and (x) preeclampsia/eclampsia.

4. Association of STH infections and being born too small for gestational age: (i) STH categories, (ii) migration status, (iii) smoking, (iv) age, (v) gravidity (primigravida vs multigravida), (vi) literacy status, (vii) underweight, (viii) short stature, (ix) GH/Chronic HTN and (x) preeclampsia/eclampsia.

Further details are provided in the Supporting Information (S1 Text).

Population-adjusted fraction (PAF) of anaemia at the first ANC contact attributable to STH infections were calculated from multivariable logistic regression models using the Stata punaf package. All other statistical analyses was performed using R software, version 3.6.1 [25].

## Results

Between July 2013 and December 2017, 83.9% (12,742/15,191) of all women registered to ANC at SMRU were included in this analysis (Fig 2). Differences in the excluded compared to

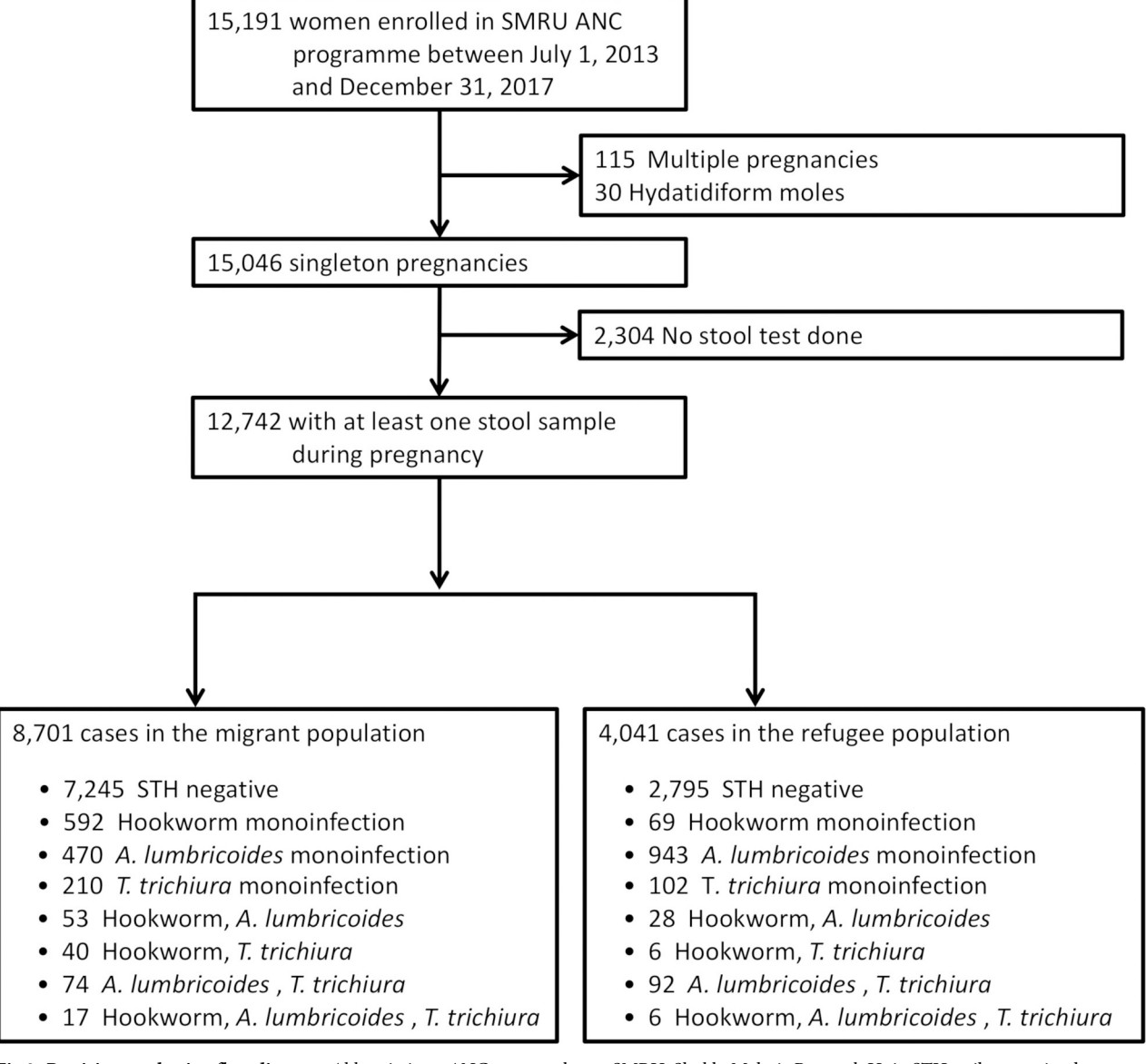

**Fig 2. Participant selection flow diagram.** Abbreviations: ANC, antenatal care; SMRU, Shoklo Malaria Research Unit; STH, soil-transmitted helminths.

the included population, e.g., a higher proportion of migrants 19.1% (2,048/10,749) than refugees 6.0% (256/4,297), multigravidae, late presenters, smokers, anaemic, and illiterate women, were expected (S1 Table).

Four in 10 women had their first ANC contact in the first trimester (39.6%, 5,052/12,742), with 46.3% (5,898/12,742) and 14.1% (1,792/12,742) in the second and third trimester, respectively. Following routine screening procedures, 93.7% (11,939/12,742) of ANC attendees provided one stool sample mostly at first contact, while multiple stool tests were done in 6.3% (803/12,742). As these procedures became routine in the second half of 2013, and as ANC operations ceased in Mae La refugee camp at the end of 2016, the total number of tests was lower in 2013 and 2017, compared to the other years (S2 Table).

Pregnant women in the refugee camp were enrolled earlier in the ANC programme than women from the migrant population; median (IQR) EGA at first ANC contact $14^{+2}$ ($9^{+0}$–$22^{+4}$) vs. $18^{+0}$ ($11^{+5}$–$24^{+6}$) weeks$^{+days}$. Consequently, STH infections were detected earlier in gestation in refugees compared to migrants (Table 1).

## Prevalence of STH infection and associated consequences

Overall, a positive stool test for STHs was identified in 21.2% (2,702/12,742; 95% confidence interval (CI) 20.5–21.9%) of pregnant women. The annualised and pooled numbers in years for which complete set of data was available for the migrant and refugee population, showed a minor reduction STH prevalence (2014, 25.7%; 2015, 21.8% and 2016: 17.8% respectively). A similar trend was observed when the prevalence was disaggregated by the migrant and refugee population (S2 Table).

STH mono-infection was the most common finding: 88.3% (2,386/2,702), while concurrent infection with two STH species was detected in 10.8% (293/2,702) and a combination of all three STH species in 0.9% (23/2,702) of women (S2 Table).

There was two-fold higher odds of STH infection in refugees with a prevalence of 30.8% (1,246/4,041) compared to migrants, with a prevalence of 16.7% (1,456/8,701). The heterogeneity of STH infection was apparent when comparing prevalence of mono-infections between these two populations, as *A. lumbricoides* was much more prevalent in the refugee camp compared to migrant clinics (23.3% (943/4,041) vs. 5.4% (470/8,701)). *T. trichiura* prevalence was similar between the migrant (2.4%, 210/8,701) and refugee population (2.5%, 102/4,041). On the contrary, a lower proportion of refugee women were infected with hookworm, 1.7% (69/4,041) compared to the migrant population, 6.8% (592/8,701). Multiple STH infections were slightly more common in the migrant population as two species were detected in 11.5% (167/1,456) of positive cases vs. 10.1% (126/1,246) in refugees, and three species in 1.2% (17/1,456) vs. 0.5% (6/1,246), respectively. STH infection was associated with eosinophilia and the association was strongest for hookworm (p<0.001), followed by *A. lumbricoides* (p<0.001), but not significant for *T. trichiura* (p = 0.493) (Table 1).

After removal of cases with missing HCT and multiple STH infections, 4.8% (480/10,035) of women with a negative stool sample were diagnosed with anaemia at the first ANC contact: migrants 4.5% (329/7,242) vs. refugees 5.4% (151/2,793). Anaemia at the first ANC contact was more common in women with STH infection compared to their uninfected counterparts: 7.1% (170/2,385) vs. 4.8% (480/10,035) (p<0.001).

*A. lumbricoides* (7.4%; 105/1,413) and hookworm (7.4%; 49/660) had an identical prevalence of anaemia at first ANC contact, while it was lower for *T. trichiura* at 5.1% (16/312) (Table 1). In an adjusted logistic regression model, the association between STH infection and anaemia at the first ANC contact remained significant for *A. lumbricoides* (adjusted odds ratio (aOR) 1.37, 95% CI 1.08–1.72) and hookworm (aOR 1.65, 95% CI 1.19–2.24), while no association for *T. trichiura* was evident (aOR 1.05, 95% CI 0.60–1.71). Overall, the intensity of infection was associated with anaemia at the first ANC contact as pregnant women with higher egg counts were more likely to present with anaemia. The strongest association was recorded for hookworm (S3 Table and S1 Fig).

As expected, age, malaria at first ANC and, in particular, late presentation to ANC were associated with an increased odds for anaemia at first ANC contact. Late presentation to ANC was also associated with an increased proportion of anaemia at the first ANC contact in STH-positive, compared to negative cases: first trimester 2.0% (19/960) STH positive vs. 1.6% (65/3,967) STH-negative; second trimester: 9.0% (101/1,129) vs. 6.5% (299/4,623); and in the third trimester 16.9% (50/296) vs. 8.0% (116/1,445). STH infection accounted for a PAF of 6.9% (95% CI 2.7–10.8%) of all anaemia at the first ANC contact, while late presentation to ANC was the strongest contributor to anaemia (S3A and S3B Table).

**Table 1. Demographic characteristics of the migrant and refugee population.** Data shown for cases with an absent STH infection in pregnancy or a monoinfection with hookworm, *A. lumbricoides* and *T. trichiura*.

| Table 1 | Migrants (n = 8,517) | | | | Refugees (n = 3,909) | | | |
|---|---|---|---|---|---|---|---|---|
| | STH neg (n = 7,245) | HW MI (n = 592) | AL MI (n = 470) | TT MI (n = 210) | STH neg (n = 2,795) | HW MI (n = 69) | AL MI (n = 943) | TT MI (n = 102) |
| EGA stool test (weeks$^{+days}$), median (IQR) | $19^{+3}$ ($12^{+4}$–$26^{+6}$) | $21^{+1}$ ($14^{+4}$–$28^{+6}$) | $21^{+4}$ ($15^{+2}$–$28^{+5}$) | $22^{+3}$ ($15^{+3}$–$28^{+3}$) | $14^{+4}$ ($9^{+0}$–$23^{+0}$) | $17^{+5}$ ($11^{+6}$–$24^{+5}$) | $14^{+4}$ ($9^{+1}$–$22^{+4}$) | $16^{+6}$ ($9^{+5}$–$25^{+1}$) |
| **Demographics** | | | | | | | | |
| Age (years), median (IQR) | 25 (21–31) | 25 (20–31) | 25 (20–30) | 23 (19–28) | 25 (21–31) | 24 (21–30) | 25 (20–31) | 23 (19–30) |
| Age group, n(%) | | | | | | | | |
| - <20 | 1,213 (16.7) | 121 (20.4) | 84 (17.9) | 54 (25.7) | 459 (16.4) | 10 (14.5) | 193 (20.5) | 27 (26.5) |
| - 20–29 | 3,790 (52.3) | 292 (49.3) | 254 (54.0) | 116 (55.2) | 1,448 (51.8) | 40 (58.0) | 447 (47.4) | 47 (46.1) |
| - 30–39 | 1,919 (26.5) | 152 (25.7) | 110 (23.4) | 33 (15.7) | 782 (28.0) | 13 (18.8) | 262 (27.8) | 24 (23.5) |
| - ≥ 40 | 323 (4.5) | 27 (4.6) | 22 (4.7) | 7 (3.3) | 106 (3.8) | 6 (8.7) | 41 (4.3) | 4 (3.9) |
| - Underweight$^{§}$, n (%) | 766 (10.6) | 77 (13.0) | 42 (8.9) | 28 (13.3) | 232 (8.3) | 7 (10.1) | 80 (8.5) | 12 (11.8) |
| Short stature¤, n (%) | 642 (8.9) | 66 (11.1) | 67 (14.3) | 21 (10.0) | 300 (10.7) | 9 (13.0) | 130 (13.8) | 11 (10.8) |
| Karen ethnicity, n (%) | 3,842 (53.0) | 258 (43.6) | 205 (43.6) | 77 (36.7) | 2,333 (83.5) | 65 (94.2) | 791 (83.9) | 92 (90.2) |
| Parity, median (IQR) | 1 (0–2) | 1 (0–3) | 1 (0–3) | 0 (0–2) | 1 (0–2) | 1 (0–3) | 1 (0–3) | 1 (0–2) |
| No. of previous pregnancies, n (%) | | | | | | | | |
| - Primigravida, n (%) | 2,515 (34.7) | 198 (33.4) | 134 (28.5) | 104 (49.5) | 913 (32.7) | 21 (30.4) | 241 (25.6) | 48 (47.1) |
| - Multigravida n (%) | 4,730 (65.3) | 394 (66.6) | 336 (71.5) | 106 (50.5) | 1,882 (67.3) | 48 (69.6) | 702 (74.4) | 54 (52.9) |
| First ANC in T1, n (%) | 2,601 (35.9) | 210 (35.5) | 127 (27.0) | 66 (31.4) | 1,370 (49.0) | 28 (40.6) | 482 (51.1) | 48 (47.1) |
| Literate, n (%) | 4,515 (62.3) | 308 (52.0) | 251 (53.4) | 129 (61.4) | 1,956 (70.0) | 24 (34.8) | 528 (56.0) | 56 (54.9) |
| Smoker, n (%) | 729 (10.1) | 79 (13.3) | 51 (10.9) | 17 (8.1) | 326 (11.7) | 16 (23.2) | 156 (16.5) | 22 (21.6) |
| Anaemia 1$^{st}$ ANC, n (%) | 329 (4.5)[i] | 38 (6.4) | 22 (4.7) | 11 (5.2) | 151 (5.4)[ii] | 11 (15.9)[iii] | 83 (8.8) | 5 (4.9) |
| Malaria in pregnancy | 158 (2.2) | 25 (4.2) | 17 (3.6) | 3 (1.4) | 18 (0.6) | 1 (1.4) | 15 (1.6) | 0 (0.0) |
| Eosinophilia (≥6%), n (%) | 815 (15.1)[1] | 137 (31.5)[2] | 65 (18.6)[3] | 25 (16.4)[4] | 234 (8.6)[5] | 10 (15.4)[6] | 129 (14.4)[7] | 11 (11.1)[8] |

Data displayed as proportions n (%) or median (IQR). Multiple infections (i.e., 184 in the migrant population and 132 in the refugee population) are not shown in Table 1.

§ defined as BMI <18.5 kg/m$^2$ according to Asian BMI groups.

¤ defined as height < 145 cm.

(i) 3 missing cases

(ii) 2 missing cases

(iii) 1 missing case.

Eosinophilia: results included if CBC taken within 28 days before the stool exam.

Eosinophilia missing cases n (%):

(1) 1,830 (25.3%) (2) 157 (26.5%) (3) 121 (25.7%) (4) 58 (27.6%) (5) 77 (2.8%) (6) 4 (5.8%) (7) 46 (4.9%) (8) 3 (2.9%).

Abbreviations: ANC, antenatal care; AL, *Ascaris lumbricoides*; BMI, body mass index; CBC, complete blood count; EGA, estimated gestational age; HW, hookworm; IQR, interquartile range; MI, monoinfection; neg, negative; STH, soil-transmitted helminths; TT, *Trichuris trichiura*; T1, first trimester (0–13 completed weeks); wt, weight (under-wt BMI<18.5 kg/m$^2$).

Malaria during pregnancy was diagnosed in 250 of 12,742 women. Of these, 91.2% (228/250) were *Plasmodium vivax* infections, 7.6% (19/250) *P. falciparum*, and 1.2% (3/250) were mixed infections. Overall, malaria was more often diagnosed in pregnant women with STH infection (2.7%, 74/2,702 vs. 1.8%, 176/10,040; p = 0.001). In an unadjusted analysis, a significant association between STH mono-infection and malaria in pregnancy was seen for hookworm (OR 2.29, 95% CI 1.51–3.49), but not for *A. lumbricoides* (OR 1.30, 95% CI 0.89–1.90) and *T. trichiura* (OR 0.54, 95% CI 0.17–1.71) when compared to STH-negative cases.

## Pregnancy outcome

The analysis of adverse pregnancy outcomes in women with a documented STH infection included: 13.8% (556 of 4,023 eligible cases) miscarriages, 1.0% (88/9,250) stillbirths, 6.1% (551/8,974) PTBs, and 18.9% (1,528/8,079) being SGA (Fig 3 and S4 Table). When adjusting

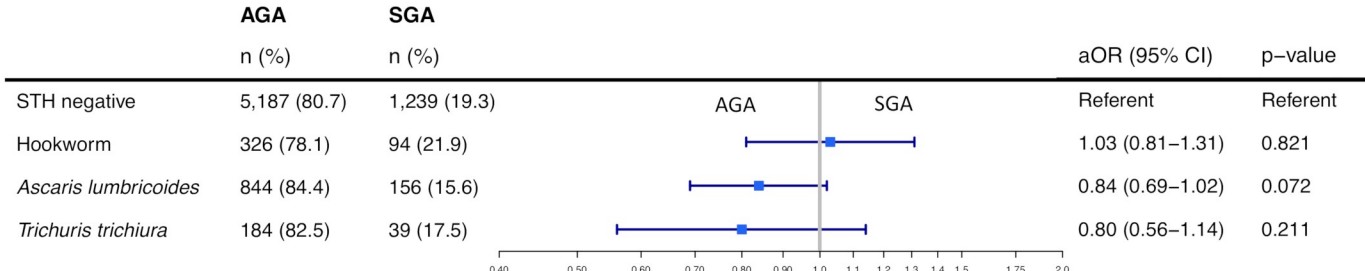

**Association between miscarriage and STHs** (Cox proportional hazards regression model)

| | Liveborn | Miscarriage | | aHR (95% CI) | p–value |
|---|---|---|---|---|---|
| | n (%) | n (%) | | | |
| STH negative | 2,738 (85.6) | 461 (14.4) | | Referent | Referent |
| Hookworm | 178 (87.7) | 25 (12.3) | | 0.74 (0.49–1.12) | 0.158 |
| *Ascaris lumbricoides* | 464 (88.7) | 59 (11.3) | | 0.63 (0.48–0.84) | **0.002** |
| *Trichuris trichiura* | 87 (88.8) | 11 (11.2) | | 0.95 (0.52–1.72) | 0.856 |

**Association between stillbirth and STHs** (Cox proportional hazards regression model)

| | Liveborn | Stillbirth | | aHR (95% CI) | p–value |
|---|---|---|---|---|---|
| | n (%) | n (%) | | | |
| STH negative | 7,314 (99.0) | 72 (1.0) | | Referent | Referent |
| Hookworm | 506 (99.2) | 4 (0.8) | | 0.64 (0.23–1.78) | 0.396 |
| *Ascaris lumbricoides* | 1,101 (99.1) | 10 (0.9) | | 0.78 (0.39–1.58) | 0.493 |
| *Trichuris trichiura* | 241 (99.2) | 2 (0.8) | | 0.78 (0.19–3.19) | 0.729 |

**Association between preterm birth and STHs** (Logistic regression model)

| | Term | Preterm | | aOR (95% CI) | p–value |
|---|---|---|---|---|---|
| | n (%) | n (%) | | | |
| STH negative | 6,722 (93.9) | 433 (6.1) | | Referent | Referent |
| Hookworm | 458 (92.0) | 40 (8.0) | | 1.38 (0.97–1.95) | 0.070 |
| *Ascaris lumbricoides* | 1,021 (94.2) | 63 (5.8) | | 0.86 (0.64–1.15) | 0.311 |
| *Trichuris trichiura* | 222 (93.7) | 15 (6.3) | | 0.86 (0.49–1.50) | 0.589 |

**Association between small for gestational age and STHs** (Logistic regression model)

| | AGA | SGA | | aOR (95% CI) | p–value |
|---|---|---|---|---|---|
| | n (%) | n (%) | | | |
| STH negative | 5,187 (80.7) | 1,239 (19.3) | | Referent | Referent |
| Hookworm | 326 (78.1) | 94 (21.9) | | 1.03 (0.81–1.31) | 0.821 |
| *Ascaris lumbricoides* | 844 (84.4) | 156 (15.6) | | 0.84 (0.69–1.02) | 0.072 |
| *Trichuris trichiura* | 184 (82.5) | 39 (17.5) | | 0.80 (0.56–1.14) | 0.211 |

**Fig 3. Forrest plots depicting the association between a soil-transmitted helminth infection and pregnancy outcomes.** Data presented as n (%) for the investigated outcomes miscarriage, stillbirth, preterm birth and being born too small for gestational age alongside the adjusted hazards ratio and odds ratio respectively. Detailed statistical methods are provided in the Supporting Information (S1 Text). Abbreviations: AGA, appropriate for gestational age; aHR, adjusted hazards ratio; aOR, adjusted odds ratio; CI, confidence interval; SGA, small for gestational age; STHs, soil-transmitted helminths.

for confounders, *A. lumbricoides* mono-infections were associated with a lower rate of miscarriage (0.63 [adjusted HR], 95% CI: 0.48–0.84, p = 0.002), with a similar, non-significant association observed for hookworm (0.74, 95% CI 0.49–1.12, p = 0.158). Non-significant associations included a higher odds of PTB with hookworm mono-infection (aOR 1.38, 95% CI 0.97–1.95, p = 0.070), and a reduced odds of SGA with A. *lumbricoides* mono-infection (aOR 0.84, 95% CI 0.69–1.02, p = 0.072). The direction of association for miscarriage and stillbirth was the same for all three STH species, while hookworm had a divergent association for PTB and SGA when compared to *A. lumbricoides* and *T. trichiura*. The intensity of infection did not have a significant impact on any of the reported pregnancy outcomes in both, unadjusted and adjusted analyses.

## Discussion

Over the past decade progress in controlling STH infections has been made; however, further effort is required to reduce the STH prevalence and attributable morbidity in primarily affected populations from low- and middle income countries in order to meet the 2030 WHO targets [26]. Currently the WHO recommends preventive chemotherapy (deworming) as a public health intervention for pregnant women, after the first trimester who live in areas where the baseline prevalence of hookworm and/or *T. trichiura* infection is 20% or higher, and the anaemia prevalence is 40% or higher among pregnant women [27]. While these criteria were not met, one in five pregnant women in this rural-tropical population at the Thailand-Myanmar border were infected with STHs. Although, the prevalence of STH infections has declined considerably over the past 20 years in this border population [28], our findings confirm that STH remain high in these vulnerable populations, in particular when considering the suboptimal stool testing method. A cross-sectional survey carried out in pregnant women from Mae La refugee camp in 1996 revealed a prevalence of STH as high as 81% (95% CI: 76–84%). In a more recent cross-sectional survey, conducted in 2007, including Mae La refugee camp and mobile migrant pregnant women, the observed STH prevalence was 62% (95% CI: 58–66%) [18], while we found a prevalence of any STH of 21.2% (95% CI: 20.5–21.9%) in samples collected between mid-2013 and the end of 2017. Importantly, the same diagnostic technique (i.e., formalin concentration method) was used in all surveys, and hence, the results are readily comparable. It should be noted, however that other techniques are available for STH diagnosis that provide higher sensitivity than formalin concentration, such as Kato-Katz thick smear or polymerase chain reaction (PCR) techniques [29].

Pregnant women from the refugee camp were two times more likely to be infected with STHs and the different distribution of STH species is suggestive of a differential risk of infection in these two populations. The *A. lumbricoides* prevalence was over four times higher in refugees, while hookworm was four times higher in migrants. The explanations for the observed heterogeneity are unclear, but sanitation and hygiene in the densely packed refugee camp may have contributed. *A. lumbricoides* eggs are particularly robust and can remain viable in the soil for over 5 years and *A. lumbricoides* is associated with high egg output [30]. Mae La refugee camp is constructed on a clay soil type that inhibits migration of the hookworm larvae, which are more suited to sandy soils [30]. Additionally, mass drug administration has been linked to a reduction of benzimidazole efficacy against *A. lumbricoides* [31] and deworming programs in the refugee population have been routine for more than 20 years. To our knowledge, no formal investigation on selective pressures and the efficacy of benzimidazole for treatment of STH infection in the area has been done.

STH infections, in particular hookworm, are associated with immediate consequences such as anaemia due to depletion of iron stores [5,9]. The prevalence of anaemia at first ANC

contact was 1.5-fold higher in pregnant women diagnosed with STH infection and the PAF suggests that STH infection account for nearly 7% of all anaemia detected at the first ANC contact. As anaemia at first ANC contact in STH-positive compared to negative cases increased from 1.2-fold to 2.1-fold from first to third trimester, early ANC attendance with detection and treatment of STH infection is an important strategy to prevent anaemia in pregnancy, and corroborates findings in published literature [13,14]. The higher eosinophil count associated with STH infection is consistent with the literature [32,33].

Hookworm infections were associated with malaria at first ANC, confirming previous epidemiologic reports from this population, albeit at a significantly declining overall malaria positivity rate over time: 27% in 1996, 20% in 2007 [18], compared with 1.8% in the current dataset. *P. vivax* was the predominant form of malaria in these populations, and the ratio of *P. vivax* to *P. falciparum* in this study is similar to the ratio found in the general population [34]. While the problem of malaria in pregnancy as an independent risk factor for adverse pregnancy outcome is well described [35,36], the role of STH infections has been largely neglected [37].

In an exploratory and adjusted analysis, having an infection with one of the three STH species was associated with a reduced risk of miscarriage, which was significant for *A. lumbricoides*. Published data on the association between *A. lumbricoides* and miscarriage is limited and anecdotal [38,39]. However, STH infections might induce immune modulation toward a non-inflammatory T-regulatory ($T_{Reg}$) and T helper 2 ($T_H2$) response, which is thought to protect the parasites from the host immune response [40–42], and thus may reduce the risk of miscarriage.

The numbers for stillbirth were low, but the absence of an association is consistent with current evidence [43]. While having a STH infection was not significantly associated with PTB or SGA, hookworm was associated with an increase in PTB while *A. lumbricoides* was associated with a decrease in SGA. Overall, the divergent associations for hookworm compared to *A. lumbricoides* and *T. trichiura*, for both of these outcomes limited the utility of pooling women infected with different species. Interestingly, the direction of associations for all adverse pregnancy outcomes were identical when only cases with a single stool test were included.

Limitations of the presented data pertain to its observational nature. Moreover, the burden of STH infections is likely underestimated, as the examination of a single stool specimen is insufficient to rule out an infection and the method used for stool examination is less sensitive when compared to other techniques [44,45]. Hence, examination techniques with increased sensitivity and specificity profiles would allow to better characterize differences between infected and uninfected groups and to more accurately estimate the infection density. As stool tests were done routinely only at ANC enrolment, any STH infection that was acquired later in pregnancy was likely missed. Easier accessibility to ANC services in the refugee camp, resulted in earlier ANC contact and detection of STHs. Moreover, women were less likely to have an unknown outcome because they could be readily traced. This difference likely increased reporting of miscarriage in the refugee population, when compared to migrants. Secondly, treatment provided routinely to women with a positive stool sample, could have potentially influenced pregnancy outcome; hence, whether STH infection treated at an EGA of 14 weeks has an effect on the risk for PTB or SGA cannot be definitively answered from the observational data reported here. Expected associations with established risk factors of anaemia and adverse pregnancy outcomes are suggestive of the robustness of the current epidemiologic data.

## Conclusion

STH are ubiquitous and infect one in five pregnant women in this vulnerable population on the Thailand-Myanmar border with considerable heterogeneity of STH infection between

refugees and migrants. Improved examination techniques (e.g., Kato-Katz methods) would provide a more accurate assessment of burden and intensity of infection and would enable better informed policy decisions. STH infections are a preventable cause of anaemia in pregnancy and, based on the findings of this analysis, systematic screening and deworming of positive cases contributes to this, as would early ANC attendance and appropriate management. The preliminary results on the association between species-specific STH infections and pregnancy outcomes–while plausible from a biological point of view–require confirmation and more in-depth investigations in controlled clinical research settings, preferably with a higher number of cases to increase the statistical power.

## Supporting information

**S1 Fig. Proportion of anaemia at the first ANC contact in relation to the infection intensity and compared between STH monoinfections.**
(DOCX)

**S1 Table. Comparison of included and excluded cases.**
(DOCX)

**S2 Table. Time trend of infected pregnant women attending the migrant and refugee ANC clinics.**
(DOCX)

**S3 Table. A Table: Associations between anaemia at first ANC contact and a STH infection in the migrant population. B Table: Associations between anaemia at first ANC contact and a STH infection in the refugee population.**
(DOCX)

**S4 Table. Pregnancy outcome compared between STH infections in migrants and refugees.**
(DOCX)

**S1 Text. Statistical methods for the estimation of the association between a soil-transmitted helminth infection and pregnancy outcome.**
(DOCX)

## Acknowledgments

We would like to thank all pregnant women who enrolled in the antenatal care program as results of this report are reliant on data generated through their participation. Additionally, we appreciate the relentless effort of all SMRU staff to improve the antenatal care for the pregnant population at the Thailand-Myanmar border.

The Shoklo Malaria Research Unit (SMRU) is part of the Wellcome Trust Mahidol University Oxford Tropical Medicine Research Programme, supported by the Wellcome Trust of Great Britain (Major Overseas Programme).

## Author Contributions

**Conceptualization:** Tobias Brummaier, Nay Win Tun, Rose McGready.

**Data curation:** Nay Win Tun, Aung Myat Min, Mary Ellen Gilder, Laypaw Archasuksan, Stephane Proux, Rose McGready.

**Formal analysis:** Tobias Brummaier, Julie A. Simpson, Rose McGready.

**Funding acquisition:** Francois Nosten.

**Investigation:** Tobias Brummaier, Nay Win Tun, Mathieu Nacher, Julie A. Simpson, Francois Nosten, Rose McGready.

**Methodology:** Tobias Brummaier, Julie A. Simpson, Rose McGready.

**Project administration:** Tobias Brummaier, Jürg Utzinger, Daniel H. Paris, Mathieu Nacher, Francois Nosten, Rose McGready.

**Resources:** Tobias Brummaier, Julie A. Simpson.

**Software:** Douwe Kiestra, Rose McGready.

**Supervision:** Prakaykaew Charunwatthana, Jürg Utzinger, Daniel H. Paris, Mathieu Nacher, Julie A. Simpson, Francois Nosten, Rose McGready.

**Validation:** Tobias Brummaier, Mathieu Nacher, Julie A. Simpson, Francois Nosten, Rose McGready.

**Visualization:** Tobias Brummaier.

**Writing – original draft:** Tobias Brummaier.

**Writing – review & editing:** Tobias Brummaier, Nay Win Tun, Aung Myat Min, Mary Ellen Gilder, Laypaw Archasuksan, Stephane Proux, Douwe Kiestra, Prakaykaew Charunwatthana, Jürg Utzinger, Daniel H. Paris, Mathieu Nacher, Julie A. Simpson, Francois Nosten, Rose McGready.

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
