## [Decision Letter · Decision Letter 0]

22 Dec 2020

Dear Mr Brummaier,

Thank you very much for submitting your manuscript "Burden of soil-transmitted helminth infection in pregnant refugees and migrants on the Thailand-Myanmar border: results from a retrospective cohort" for consideration at PLOS Neglected Tropical Diseases. As with all papers reviewed by the journal, your manuscript was reviewed by members of the editorial board and by several independent reviewers. In light of the reviews (below this email), we would like to invite the resubmission of a significantly-revised version that takes into account the reviewers' comments. 

We cannot make any decision about publication until we have seen the revised manuscript and your response to the reviewers' comments. Your revised manuscript is also likely to be sent to reviewers for further evaluation.

Sincerely,

Keke Fairfax, PhD

Deputy Editor

Keke Fairfax

Deputy Editor

Reviewer's Responses to Questions

**Key Review Criteria Required for Acceptance?**

**Methods**

-Are the objectives of the study clearly articulated with a clear testable hypothesis stated?

-Is the study design appropriate to address the stated objectives?

-Is the population clearly described and appropriate for the hypothesis being tested?

-Is the sample size sufficient to ensure adequate power to address the hypothesis being tested?

-Were correct statistical analysis used to support conclusions?

-Are there concerns about ethical or regulatory requirements being met?

Reviewer #1: see general comments

Reviewer #2: Given the primary objectives of the study, it was expected that the evidence would be used to inform health policy and practice in the two study populations of migrants and refugees in terms of timing and frequency of antenatal care and public health interventions like deworming and iron supplementation. Given also the significant difference in the occurrence of STH infection in the two study populations, an appreciation of what is currently being provided in terms of antenatal care in the respective settings would be warranted and, if necessary, appropriate population-specific remedial action proposed. An analysis of relevant outcomes, again in the two populations separately, would seem the most beneficial in terms of subsequent health interventions that should be considered by health authorities. Therefore all analyses should, as a minimum, be disaggregated by population.

The rationale for undertaking exploratory analyses is missing. For example, an examination of the association between STH and anemia is not informative, as this association is well known and one would design a different study to examine such an association anyway (e.g. STH diagnosis would be based on a different test). The rationale for all exploratory analyses therefore should be clarified or the analyses should be removed. 

Retrospective data can certainly be used to explore the study’s primary objectives. But there are limitations to this type of data. Selection bias is one such limitation. Selection bias here can occur from missing important numbers of pregnant women in the database, or limiting the numbers of women in certain analyses because of incomplete data. To give one example, as ANC was known to only be fully operational at the end of 2013 in the study populations until the end of 2016, why were data to be analyzed not restricted to the three full years of 2014, 2015 and 2016? It should be kept in mind that the ‘prevalence’ reported in this study is a 54-month cumulative prevalence. It would be helpful to annualize this figure. It might then be clarified whether STH is, or is not, of public health importance in these populations and that relevant health policy and practice should be reviewed. Also, as migrant and refugee populations were found to be significantly different in terms of STH infection (despite the measurement error), all analyses should be performed for each of these study populations separately.

The question of trimester is always problematic. WHO defines the first 16 weeks of pregnancy as the first trimester, so why was 14 weeks used in this study (line 200)? (WHO. Integrated Management of Pregnancy and Childbirth. Pregnancy, Childbirth, Postpartum and Newborn Care: A guide for Essential Practice. WHO, 2015; see page 180.) At a minimum, the data need to re-analyzed with the WHO definition of trimester. The definition of trimester is important as this will affect the estimate of the number of miscarriages (line 200). All instances referring to trimester in the text and tables require revision.

The measurement of the STH infection itself is problematic and subject to measurement error. WHO recommends that the Kato-Katz method be used to assess the prevalence and intensity of STH infection in order to then implement a preventive chemotherapy program in endemic areas. As a non-standardized diagnostic test was used in this study, with unknown estimates of species-specific sensitivity and specificity, it is difficult to interpret the results. Therefore, the accuracy of the estimates of prevalence and intensity for monoinfections, or the prevalence of multiple species infections, is unknown, and any analysis using these figures is subject to measurement error. This is one of the reasons for dropping the exploratory analyses. It is unclear why the 6.3% due to multiple stool testing from the same individuals was not removed from all analyses…They bias the results if they are not removed.

Reviewer #3: The study population, objectives, design and ethics are well described and appropriate statistical analyses are employed to support their conclusions.

**Results**

-Does the analysis presented match the analysis plan?

-Are the results clearly and completely presented?

-Are the figures (Tables, Images) of sufficient quality for clarity?

Reviewer #1: see general comments

Reviewer #2: All results tables, figures and supplementary material need to be reconsidered given the above comments.

Reviewer #3: Results are clearly presented in line with the study objectives. However, I identified a few issues below, these need to be addressed

1. A couple of numbers in the text do not match what is indicated in the tables, let the authors be consistent with this. 

2. Lines 256 – 258: STH infections were detected earlier… The numbers on line 257 e.g. 14+2(9+0 – 22+4) do not match the numbers in Table 1 

3. Table S2: what is the motivation for showing STH neg while for the other mono infections it shows positivity? Wouldn’t it be better if all infections showed the same thing (either positivity or negativity)?

4. Lines 269 – 281: The denominators are 4041 and 8701 for refugee and migrant populations respectively. This contradicts the figures in Table 1 i.e. 3909 and 8517 respectively. Please ensure that this is well explained.

5. Line 289: In an adjusted logistic regression model… I think that the potential confounders adjusted for need to be mentioned, perhaps in the methods section.

6. Lines 293 – 294: would you consider a trend test to confirm the result that the intensity of infection was associated with anaemia?

7. Lines 307 – 309: in an unadjusted analysis… you have reported adjusted analyses before, why not for this result also?

8. Please make Figure 3 clearer, it looks fuzzy at the moment

9. Lines 315 – 316: when adjusting for confounders, … please mention these confounders in the methods section. I know they appear in supplementary information but it might be better to include them in the main text.

**Conclusions**

-Are the conclusions supported by the data presented?

-Are the limitations of analysis clearly described?

-Do the authors discuss how these data can be helpful to advance our understanding of the topic under study?

-Is public health relevance addressed?

Reviewer #1: see general comments

Reviewer #2: The discussion also then should be focussed on the primary objectives to inform public health interventions to the two study populations. There is considerable repetition of results in the discussion which can be removed. All mention of benzimidazole efficacy should be removed as this was not adequately investigated in this study.

Reviewer #3: Conclusions are supported by the data and limitations are largely well described, save for the statistical power for association analyses.

Public health relevance is addressed, though I think a bit more of this needs to appear in the abstract!

**Editorial and Data Presentation Modifications?**

Reviewer #1: see general comments

Reviewer #2: (No Response)

Reviewer #3: (No Response)

**Summary and General Comments**

Reviewer #1: Burden of soil-transmitted helminth infection in pregnant refugees and migrants on the Thailand-Myanmar border: results from a retrospective cohort.

By Brummaier et al

General comment

The article is well written and the experiment is well conducted but in my opinion this paper has an important drawback: 

The authors do not recognize that the women population selected presents low level of STH infection (around 20%) principally due to A lumbricoides and therefore their experiment had no possibility to show any morbidity caused by the parasites.

A population with so low STH prevalence of hookworm (1%-6%) and T. trichiura (2.4%) do not qualify for preventive chemotherapy according the WHO recommendations 

(please refers to WHO guidelines 2017:“ Preventive chemotherapy, using single-dose albendazole (400 mg) or mebendazole (500 mg), is recommended as a public health intervention for pregnant women, after the first trimester living in areas where both: (i) the baseline prevalence of hookworm and/or T. trichiura infection is 20% or higher among pregnant women, and (ii) anaemia is a severe public health problem, with a prevalence of 40% or higher among pregnant women.”)

The reason why PC is not recommended in populations of women where the prevalence of Hookworm and T. trichiura is lower than 20% is because at this level of prevalence the only infections present are the ones of light intensity that are not expected to cause morbidity. 

I think that an experiment conducted in this population with so low pevalence had no possibility to demonstrate any morbidity caused by the parasites and therefore is probably unnecessary.

However, there is a specific reason why suggest to reject this article despite the fact that I considered well written and resulting from a well conducted experiment: 

Unfortunately, researchers that conduct systematic review are frequently not experienced on the specific disease but only on the methodology for conducting the review; as consequence, especially in the area of deworming, when the results of studies similar to the ones I am reviewing here (no possibility to demonstrate an effect because of the low prevalence) are summarized with the ones conducted in population where the level of parasites warrant intervention, they produce serious distortion of the results of the systematic review and therefore are leading to inapropriate conclusion. 

This is why I consider this kind of articles not only unnecessary but counterproductive and therefore I reccomend rejection.

Reviewer #2: This manuscript details a retrospective review of cumulative data (recorded between 2013 and 2017) from screening for STH infection among migrant and refugee pregnant women living in camps along the Thai-Myanmar border. It is actually unclear why the study’s objective was to describe and compare STH infection between these two study populations. Describe yes, but why compare? Would results influence screening procedures or deworming policy/practice? If this was the intent, it needs to be made clearer.

Reviewer #3: This is a very well-written paper about the burden of STH infection in a pregnant population living on the Thailand-Myanmar border. Authors compare the prevalence of different STH species between a migrant population and a population of refugees in a camp, they also report an exploratory analysis on associations between STH infection, maternal health and pregnancy outcomes.

I think though, that the authors need to acknowledge the limitation of lack of statistical power for the exploratory association analyses or else confirm that this is not a problem.

PLOS authors have the option to publish the peer review history of their article (what does this mean?). If published, this will include your full peer review and any attached files.

Reviewer #1: No

Reviewer #2: No

Reviewer #3: Yes: Lawrence Lubyayi
---

## [Editor Report · Decision Letter 1]

8 Feb 2021

Dear Mr Brummaier,

We are pleased to inform you that your manuscript 'Burden of soil-transmitted helminth infection in pregnant refugees and migrants on the Thailand-Myanmar border: results from a retrospective cohort' has been provisionally accepted for publication in PLOS Neglected Tropical Diseases.

Best regards,

Keke C Fairfax, PhD

Deputy Editor

Keke Fairfax

Deputy Editor

---

## [Editor Report · Acceptance letter]

23 Feb 2021

Dear Mr Brummaier,

We are delighted to inform you that your manuscript, "Burden of soil-transmitted helminth infection in pregnant refugees and migrants on the Thailand-Myanmar border: results from a retrospective cohort," has been formally accepted for publication in PLOS Neglected Tropical Diseases.

Best regards,

Shaden Kamhawi

co-Editor-in-Chief

Paul Brindley

co-Editor-in-Chief
